# HSP90 Inhibitor, 17-DMAG, Alone and in Combination with Lapatinib Attenuates Acquired Lapatinib-Resistance in ER-positive, HER2-Overexpressing Breast Cancer Cell Line

**DOI:** 10.3390/cancers12092630

**Published:** 2020-09-15

**Authors:** Hye Jin Lee, Seungho Shin, Jinho Kang, Ki-Cheol Han, Yeul Hong Kim, Jeoung-Won Bae, Kyong Hwa Park

**Affiliations:** 1Division of Oncology/Hematology, Department of Internal Medicine, Korea University College Medicine, Seoul 02841, Korea; lhj5727@korea.ac.kr (H.J.L.); kang701@korea.ac.kr (J.K.); yhk0215@korea.ac.kr (Y.H.K.); 2Departments of Breast and Endocrine surgery, Korea University College Medicine, Seoul 02841, Korea; sshhere@korea.ac.kr (S.S.); kujwbae@korea.ac.kr (J.-W.B.); 3Center for Theragnosis, Biomedical Research Institute, Korea Institute of Science and Technology, Seoul 02456, Korea; biolord@kist.re.kr

**Keywords:** breast cancer, ER (+) HER2 (+), lapatinib resistance, HSP90

## Abstract

**Simple Summary:**

Lapatinib is a tyrosine kinase inhibitor widely used as a treatment for a Human Epidermal growth factor Receptor 2 (HER2) (+) breast cancer patients. However, when resistance is acquired through continued exposure, and it is associated with a poor prognosis for patients. In this study, we identified HSP90 as a common node for acquired resistance to lapatinib in two lapatinib resistant cell lines using proteomic analysis. Notably, in vitro and in vivo studies demonstrated synergy effect between lapatinib and an HSP90 inhibitor were observed in the estrogen receptor (+) HER2 (+) breast cancer cell only. These results could be a potential strategy for future clinical trials for HSP90 inhibitors in treatment—refractory HER2 (+) metastatic cancer patients

**Abstract:**

Lapatinib, a Human Epidermal growth factor Receptor 2 (HER2)-targeting therapy in HER2-overexpressing breast cancer, has been widely used clinically, but the prognosis is still poor because most patients acquire resistance. Therefore, we investigated mechanisms related to lapatinib resistance to evaluate new therapeutic targets that may overcome resistance. Lapatinib-resistant cell lines were established using SKBR3 and BT474 cells. We evaluated cell viability and cell signal changes, gene expression and protein changes. In the xenograft mouse model, anti-tumor effects were evaluated using drugs. Analysis of the protein interaction network in two resistant cell lines with different lapatinib resistance mechanisms showed that HSP90 protein was commonly increased. When Heat Shock Protein 90 (HSP90) inhibitors were administered alone to both resistant cell lines, cell proliferation and protein expression were effectively inhibited. However, inhibition of cell proliferation and protein expression with a combination of lapatinib and HSP90 inhibitors showed a more synergistic effect in the LR-BT474 cell line than the LR-SKBR3 cell line, and the same result was exhibited with the xenograft model. These results suggest that HSP90 inhibitors in patients with lapatinib-resistant Estrogen Receptor (ER) (+) HER2 (+) breast cancer are promising therapeutics for future clinical trials.

## 1. Introduction

Breast cancer is known as the most frequently diagnosed cancer in women worldwide and one of the main causes of death. In particular, 20 to 30% of breast cancers are classified as human epidermal growth factor 2 (HER2) positive and have aggressive biological characteristics compared to other breast cancer subtypes, reducing overall survival [1,2]. HER2/Neu (ErbB2) is a member of the ErbB family of transmembrane receptor tyrosine kinases, which also includes the epidermal growth factor receptor (EGFR, ErbB1), HER3 (ErbB3), and HER4 (ErbB4). In particular, co-expression of human epidermal growth factor receptor 2 (HER2) and EGFR has been reported to be associated with worse survival in HER2-positive breast cancer patients. Activation of the HER2 pathway leads to regulate cellular processes such as proliferation, differentiation, motility, and adhesion. Downstream pathwayare activated by this pathway include PLC-γ1, MAPK/Erk1/2, and PI3K/Akt, Src, the stress-activated protein kinases (SAPKs), PAK-JNKK-JNK, and the signal transducers and activators of transcription (STATs) [3,4,5]. Fortunately, advancements over the last two decades in the development of HER2-targeted agents and their clinical application has revolutionized the prognosis of patients with HER2 (+) cancer; median progression-free survival in the 1st line setting has been extended to 18.7 months with dual-target agents from 4.6 months with chemotherapy only [6].

Now, HER2 (+) breast cancers are considered highly sensitive to chemotherapy and/or targeted agents. However, with continued exposure to therapeutics, cancer cells adaptively activate alternative survival pathways and reprogram tumor biology. However, efforts for new drug development other than HER2-directed agents are insufficient. Lapatinib is a dual inhibitor targeting both HER2 and EGFR that has an anti-cancer effect on HER2-positive breast cancer by inhibiting MAPK/Erk1/2 and PI3K/Akt downstream pathways [7]. Currently, a combination of lapatinib and capecitabine has been approved for use in patients with advanced HER2-positive breast cancer after treatment with chemotherapy using dual anti-HER2 therapy; trastuzumab and pertuzumab [8]. Despite the proven clinical benefit, most patients eventually acquire resistance to lapatinib therapy in less than 9 months [8]. Evidence suggests that lapatinib resistance occurs through several mechanisms leading to activation of alternative survival pathways, such as EGFR, ERBB3, ERBB4, EPHA2, IGF1R, MET, MERTK, PIK3CA mutations, ER, and β1-integrin signaling [9,10,11,12,13,14,15,16,17,18]. Although various resistance mechanisms have been studied, the mechanism for acquired lapatinib resistance that can be exploited for next step treatment has not been clearly elucidated.

This study identified the mechanism of resistance in lapatinib-resistant breast cancer cell lines using proteomics and examined whether the identified mechanisms overcome lapatinib resistance through in vitro and in vivo models. Our data suggest that HSP90 is a target for lapatinib-resistant cancers, but ER-positive subtypes may be a more suitable population in future clinical development.

## 2. Results

### 2.1. Establishment of Lapatinib-Resistant Cell Lines In Vitro

HER2-overexpressing SKBR3 and BT474 cells were exposed to increasing concentrations of lapatinib for 1 year. Lapatinib-resistant cell lines were mentioned as LR-SKBR3 and LR-BT474, respectively. Sensitivities of parent cells and long term lapatinib-exposed cells were assessed by MTT assays to validate acquired resistance. The IC_50_ for SKBR3 and BT474 cells were 52 nM and 30 nM, respectively. In comparison, those of LR-SKBR3 and LR-BT474 were 4.5 μM and 0.7 μM, respectively (Figure 1A).

To identify alterations in tyrosine kinase signals in lapatinib-resistant cell lines, deregulation status was assessed by western blot analysis (Figure 1B, Appendix A). In LR-SKBR3 cells, phosphorylated forms of EGFR, HER2, and ERK were still active, as shown in parent cells. Treatment of LR-SKBR3 with increasing concentrations of lapatinib was effective in suppressing p-EGFR and p-HER2, but not p-Akt and p-ERK. Conversely, the suppressive activity of lapatinib for signals through p-EGFR, p-HER2, p-Akt, and p-ERK in BT474 cells were reversed in LR-BT474 cells. Thus, phosphorylated forms of EGFR and HER2 were increased with increasing concentrations of lapatinib in LR-BT474 cells, suggesting that continued exposure of resistant cells to lapatinib may induce rebound activation of targets in BT474 cells. 

### 2.2. Gene Set Enrichment Analysis of Gene Expression in Lapatinib-Resistant Breast Cell Lines 

Microarray analysis was performed to compare gene expression between parent and lapatinib-resistant breast cancer cell lines. Gene set enrichment analysis (GSEA) revealed that LR-SKBR3 cells were significantly enriched with “G2M checkpoint” and “E2F target” genes compared with the SKBR3 cell line (Figure 2A). Conversely, “early estrogen response” and “late estrogen response” genes were significantly enriched in the LR-BT474 cell line compared to the parent cell line (Figure 2B). These data indicate that the major mechanisms involved in acquired lapatinib resistance in the two cell lines are different, with an ER-related mechanism in the BT474 cell line and cell cycle-related mechanisms in the SKBR3 cell line. Further analyses were performed to investigate differences in mechanisms contributing to the development of lapatinib resistance of LR-SKBR3 and LR-BT474 cell lines. Transcriptomic profiles demonstrated that LR-SKBR3 cells were significantly enriched for processes associated with “TNF-α signaling,” “Hypoxia,” “Interferon-alpha response,” and “interferon-gamma response” compared to LR-BT474 cells (Figure 2C). In contrast, LR-BT474 cells showed significant enrichment for genes associated with “early estrogen response,” “late estrogen response,” “DNA repair,” and “apoptosis” compared to the LR-SKBR3 cell line (Figure 2D). “Early estrogen response” refers to genes whose expression is increased in 3 to 4 h, and “late estrogen response” refers to genes whose expression is increased in 24 h. Therefore, lapatinib resistance was acquired through different mechanisms for ER (+), HER2 (+) LR-BT474 and ER (−), HER2 (+) LR-SKBR3 cell lines. 

### 2.3. Phosphoproteomic Analysis Reveals HSP90 as Potential Regulator of Lapatinib Resistance 

To define acquired resistance to lapatinib in HER2 (+) breast cancer, SILAC-based quantitative phosphoproteomics were performed in lapatinib-sensitive and lapatinib- resistant cell lines (Figure 3A). A total of 3796 phosphopeptides from 1954 proteins obtained from SKBR3 and 4103 phosphopeptides from 1995 proteins obtained from BT474 were analyzed. 

Pathway and gene ontology (GO) analyses were performed with all proteins obtained from differentially upregulated phosphopeptides (>2-fold) using Enrichr. Spliceosome, mRNA surveillance pathway, and RNA transport were upregulated in LR-SKBR3, while RNA transport, and AMPK signaling pathway, mTOR signaling pathway were enriched in LR-BT474 (Figure 3B,C). These results indicate that lapatinib altered cellular pathways differently for the 2 cell lines. Next, protein-protein interaction networks were analyzed using STRING with medium confidence score (0.4) [15]. Networks were further characterized using the Cytoscape plugin clustermaker and GLay community clustering. The biological processes in the major cluster of SKBR3 were diverse but included histone H3 deacetylation, chromatin remodeling, and positive regulation of macroautophagy, while negative regulation of mTOR signaling cascade, translation initiation, and regulation of translational initiation were included in the BT474 cluster. Interestingly, the mTOR signaling pathway was enriched in LR-BT474, but several genes related to negative regulation of mTOR signaling were present in the major cluster of BT474, including AKT1S1, DEPTOR, PRKAA1, and TSC2 (Figure 3D,E). In addition, heat shock protein 90 alpha family class A member 1 (HSP90AA1) was a major protein interacting with other proteins in both clusters, and phosphorylation at Ser263 of HSP90AA1 was upregulated in both LR-SKBR3 and LR-BT474. 

### 2.4. Effect of HSP90 Inhibitor in Lapatinib-Resistant Cell Lines

The HSP90 chaperone complex regulates the stability, activation, and maturation of more than 200 oncogenic client proteins. So, targeting HSP90 offers the potential to simultaneously destroy many proteins involved in the lapatinib resistance mechanism [19,20].

To determine the effect of HSP90 inhibition, cell proliferation was evaluated after treatment with a HSP90 inhibitor, 17-dimethylaminoethylamino-17-demethoxygeldanamycin (17-DMAG) in HER2 (+), lapatinib-sensitive, and lapatinib-resistant cell lines. 17-DMAG treatment was effective in inhibiting cell proliferation of both lapatinib-resistant and parent cell lines in a dose-dependent manner. As shown in Figure 4A, the IC_50_ values of lapatinib-resistant cell lines (619 nM for LR-SKBR3, 103 nM for LR-BT474) were significantly higher than those of parent cells (24 nM for SKBR3, 16 nM for BT474).

HSP90 inhibition still induced downregulation of HER2, EGFR, and Akt to the same extent in resistant cells compared to parent cells (Figure 4B, Appendix A). Dephosphorylation of those proteins was concordant with decreases in total forms. Of note, phosphorylated forms of ERK also decreased with increasing concentrations of 17-DMAG without any change in total form, as previously reported [21].

### 2.5. Combination of 17-DMAG and Lapatinib Is Synergistic in LR-BT474 Cell

17-DMAG and lapatinib in each cell line were used to assess synergy of the two drugs in a dose-dependent manner. Synergy studies were done with the doses of lapatinib and 17-DMAG determined in the MTT assays using a single drug (Figure 1 and Figure 4). Sub-IC50 doses in each parent cells were used for initial combination. In the LR-BT474 cell line, combinations of lapatinib 50 nM + 17-DMAG 50 nM, lapatinib 100 nM + 17-DMAG 100 nM, and lapatinib 500 nM + 17-DMAG 500 nM had CI values < 1, suggesting synergism. However, LR-SKBR3 cells showed an additive effect (CI = 1) rather than synergism (C < 1) in most combinations (Figure 5A). To further evaluate the effects of 17-DMAG and lapatinib combination on downstream pathways in parents and lapatinib-resistant cell lines, total and phosphorylated forms of previously deregulated proteins were assessed (Figure 5B, Appendix A). Western blot analysis showed that a combination of lapatinib and 17-DMAG was synergistic in suppressing phosphorylated forms of HER2, EGFR, Akt, and ERK in LR-BT474 cells. However, there was no more efficacy than from single treatment in LR-SKBR3 lines. These results indicate that inhibition of HSP90 can reverse lapatinib resistance in ER (+) and HER2 (+) cancers only. 

### 2.6. Antitumor Efficacy of 17-DMAG and Lapatinib Combination in Xenograft Model

In vivo antitumor effects of 17-DMAG and lapatinib combinations were investigated in a xenograft model. Implanted tumors were established with parent cells and lapatinib-resistant cells on each flank. Tumor volume growth and inhibition after each treatment from 15 to 53 days after tumor implantation are plotted in Figure 6. At day 53, BT474 tumor-bearing mice treated with vehicle only (control) reached a mean tumor volume of 1078 ± 198 mm^3^. In contrast, the mean tumor volume of groups treated with 17-DMAG only, lapatinib only, or a combination of 17-DMAG and lapatinib at day 53 were 727 ± 94 mm^3^, 738 ± 127 mm^3^, and 435 ± 112 mm^3^, respectively. A significant additive effect in tumor growth inhibition was observed with the combination of 17-DMAG and lapatinib.

In contrast, LR-BT474 tumor-bearing mice showed similar tumor volume among the experimental groups of control, lapatinib only, or 17-DMAG only (Figure 6B). In the group with combination of 17-DMAG and lapatinib, tumor volume at day 51 was 633 ± 116 mm^3^ and it showed significant synergism with growth inhibition. After sacrifice of BT474 and LR-BT474 tumor-bearing mice, the expression level of HER2 and ER in tumors was examined by immunohistochemistry. In BT474 mice, HER2 expression (3+) and ER (Allred score 6–7) expression levels showed little difference in the group with combination 17-DMAG and lapatinib compared to vehicle (control). However, in LR-BT474 mice, HER2 expression (2–3 +) was slightly reduced in the group with combination 17-DMAG and lapatinib compared to vehicle (control), and ER (Allred score 6–8) expression was considerably reduced (Figure 6C).

The same experiment was tried using SKBR3 and LR-SKBR3, but implanted tumor establishment failed with several attempts. Taken together, in vivo data confirmed our in vitro results to suggest that combination of 17-DMAG is a potential target to overcome lapatinib resistance in BT474 cells.

## 3. Discussion 

The paradigm of treatment for HER2 (+) breast cancer has evolved from a cytotoxic chemotherapy plus single antibody combination to a dual-targeted strategy and antibody-drug conjugates [22,23]. Recently, the role of HER2 tyrosine kinase inhibitors (TKIs) has been expanded in clinical practice for heavily pre-treated patients with extensive acquired resistance mechanisms [24]. However, scientific evidence to support the development of next step treatment targeting mechanisms other than HER2 remain insufficient. Of the several proposed mechanisms, HSP90 is considered a promising target, but no drugs have been approved for clinical application yet.

In this study, we identified that HER2 (+) breast cancer cells have different molecular mechanisms of acquired lapatinib resistance according to ER positivity at the transcriptional level. Further proteomic analysis confirmed HSP90 as an important node for acquired resistance; thus, HSP90 is a potential target for both LR-SKBR3 and LR-BT474 cells. In vitro data, however, demonstrated that combination of lapatinib and HSP90 inhibitor (17-DMAG) were synergistic in LR-BT474 only. A xenograft model showed clear synergy in suppressing LR-BT474 cells by combination lapatinib and 17-DMAG. Taken together, our results indicate that acquired resistance to lapatinib develops from different mechanisms according to ER positivity, and HSP90 can be targeted in ER (+) BT-474 cells only.

The HSP90s are highly conserved molecular chaperone proteins as one of the sub-family of HSPs and have distinctive role in stability and function of conformationally labile proteins. In cancer cells, HSP90 expression is induced by microenvironmental stresses, including hypoxia, chemotherapy, and immunologic attack, rather than gene amplification or point mutation [25]. It’s chaperone complex regulates the stability, activation, and maturation of more than 400 client proteins, including oncogenic receptor tyrosine kinases (HER2, EGFR, IGF-1R, MET), signaling proteins (AKT, ERK), and cell cycle regulatory proteins. For the reasons above, HSP90 has been highlighted as a new target for cancer therapy, and significant efforts have been made to develop HSP90-targeting drugs as a new option for standard of care in metastatic cancers. Due to toxicities and limited efficacy, first generation HSP90 inhibitors were not further developed for clinical applications [26]. Failure of the first-generation agents was mostly attributed to insufficient dosing and resultant inadequate target inhibition. A new generation of HSP90 inhibitors with improved toxicities are under active clinical investigation as single drugs or in combination with other therapeutics in patients with various cancers, including lung [27,28], brain [29], and breast [30,31] cancers as well as sarcomas [32,33]. Docetaxel is being used as a first-line combination therapy for patients with HER-2 positive metastatic breast cancer [34]. Of note, in non-small cell lung cancers (NSCLC), the single agent ganetespib showed promising activity in patients with tumors of oncogenic addiction to EGFR mutation or EML4-ALK rearrangements [28]. In contrast, the addition of genetespib to docetaxel showed no additional clinical benefit in a phase 3 trial involving patients with adenocarcinomas and wild-type EGFR [35]. Collectively, current evidence shows that single-agent activity was not successful in early phase trials, even in heavily pre-treated patients with active adaptive stress responses, which are chaperones for HSP90. Rather, clinical trial results indicate that HSP90 inhibitors might be an alternative strategy for salvage treatment in tumor types addicted to driver genetic alterations. Most of the switched survival signals and key nodes of pathways are client proteins of HSP90.

In breast cancer, HER2 is an important client protein of HSP90. Most of the switched survival signals and key nodes of pathways after resistance are also client proteins of HSP90 [36]. As expected, first-generation inhibitors showed strong growth inhibition in pre-clinical studies [37,38]. Most of the clinical evaluations using HSP90 inhibitors have been conducted in previously treated HER2 (+) cancers as single drugs or in combination with trastuzumab. In combination with trastuzumab, the clinical efficacy of first- and second-generation geldanamycin-based agents were modest, and the response rate was 22% at most [39,40,41]. As an example, ganetespib, a second generation non-geldanamycin based small molecule, showed 2 partial responses (PRs) in a trastuzumab-refractory ER+/HER2+ MBC (*n* = 13) cohort [30] in the first clinical trial, and triple combination paclitaxel, trastuzumab, and ganetespib demonstrated a response rate of 22% in the subsequent phase I trial [31]. Considering their modest efficacy and the current situation with more available antibody-drug conjugates and tyrosine kinase inhibitors [22,24], HSP90 inhibitors are not an attractive option for HER2 (+) metastatic cancers. Nevertheless, as all patients eventually experience treatment resistance, research into biomarker discovery in patients who showed objective responses should continue.

In HER2 (+) metastatic breast cancers, until recently, most clinical trials were conducted with a one-size-fits-all approach, regardless of ER expression. The first implication was noted from preclinical data using a lapatinib resistance model in ER^+^/HER2^+^ MCF7 HER2-18 cells and clinical tumor samples [11]. Consistent with our data, ER (+) HER2 (+) tumor cells and clinical samples showed increased ER expression with acquired resistance to lapatinib. In the MONARCHER trial, investigators showed superior progression free survival with combination of fulvestrant, trastuzumab, and abemaciclib compared to trastuzumab plus standard-of-care chemotherapy in patients with treatment-resistant ER (+) HER2 (+) metastatic breast cancers [42]. This finding proved that co-targeting ER and HER2 is a better strategy than continued HER2 inhibition with chemotherapy in the ER (+) sub-population. Similarly, the current study showed a clear difference in mechanisms for acquired resistance for SKBR3 and BT474 cells. Long-term exposure to lapatinib induces total forms of HSP90 client proteins (EGFR and HER2) in SKBR3 cells only (Figure 1B). Compared with parent cells at the transcriptional level, GSEA shows that E2F targets and G2M checkpoints are enriched in LR-SKBR3 cells, but genes for early and late estrogen responses are mostly enriched in LR-BT474 cells. Compared with the two resistant cell lines, LR-BT474 cells are significantly more enriched for genes involved in early and late estrogen responses, DNA repair, and apoptosis than LR-SKBR3 cells (Figure 2C). Conversely, genes for TNFα, interferon γ, interferon α, and hypoxia were more significantly enriched in LR-SKBR3 cells than LR-BT474 cells (Figure 2D). Thus, although HSP90 was identified as a common target in both resistant cells in our study, the target chaperones affected by HSP90 inhibition might differ. 

A synergistic effect between lapatinib and HSP90 inhibitor was also demonstrated in both parent cells of BT474 and SKBR3 in a previous study [43], consistent with our study. In lapatinib-resistant cells, however, synergy was observed in LR-BT474 cells only (Figure 5A). The in vivo anti-tumor effect was also consistent (Figure 6). Given the data from GSEA and IHC analysis of the in vivo experiment, down-regulation of ER expression in LR-BT474 cells might be an important role in the synergistic effect of 17-DMAG. As for TNFα, interferon γ, interferon α, and hypoxia, more enriched resistant mechanisms in the LR-SKBR3 cells do not seem to have much effect by 17-DMAG. Besides ER expression, better efficacy in LR-BT474 cells might be from the fact BT474 may have more deregulated targets for HSP90 inhibitor due to mutated *PIK3CA*. Although previous clinical studies did not include precise genomic information, the data presented here partially explain the significant gap with ganetespib between promising pre-clinical activity [43] and modest clinical efficacy in metastatic HER2 (+) cancers [31]. In a preclinical and clinical study using PU-H71, 5 of 6 ER (+) HER2 (+) primary breast cancers showed potential for sensitivity to the drug [38]. Of note, in clinical trials using single-agent HSP90 inhibitors, all patients who showed partial responses had ER (+) HER2 (+) breast cancers [30,44]. Together, our data and pre-clinical and clinical findings indicate that a combination strategy of HSP90 inhibitors with HER2-targeted agents may have more clinical benefits in patients with ER (+) HER2 (+) subtype breast cancer.

Lapatinib-resistant HER2 (+) breast cancer cells were used in this preclinical model because the biology of these cells may represent tumors in patients who have resistant tumors after using available HER2-targeting agents, including lapatinib. The next-generation HER2 tyrosine kinase inhibitors, neratinib, and tucatinib are now approved and available for clinical use. Although there is an ongoing study of protein changes after short-term exposure to 3 different HER2 TKIs, there is no comparative study on resistance mechanisms by TKIs. Further research is needed to determine whether the results of this study will be applied to patients who have resistance to newly developed HER2-targeted agents. 

## 4. Materials and Methods

### 4.1. Cell cultures and Reagents

The BT474 cell line was maintained in Dulbecco modified Eagle medium (DMEM) supplemented with 10% fetal bovine serum (FBS) and 1% penicillin/streptomycin. The SKBR3 cell line was maintained RPMI1640 supplemented with 10% FBS and 1% penicillin/streptomycin. All cell lines were cultured in humidified incubation at 37 °C with 5% CO_2_. Lapatinib was purchased from Santa Cruz (Dallas, TX, USA), and 17DMAG-HCL was purchased from Selleck Chemicals (Houston, TX, USA). Stock solutions were prepared with dimethyl sulfoxide (DMSO) and stored at −20 °C. Lapatinib and 17DMAG for animal experiments were purchased from LC laboratories. Lapatinib was dissolved in water with 0.5% carboxymethylcellulose, 1.8% sodium chloride, and 0.4% Tween 80. 17 DAMG was dissolved in water with 0.8% NaCl.

### 4.2. Selection of Lapatinib-Resistant Breast Cancer Cell Lines

Lapatinib-resistant cells were established by gradually increasing the concentration of lapatinib and exposing cells repeatedly. SKBR3 and BT474 cells were initially exposed to 50 nM concentrations of lapatinib for 48 h in medium containing 10% FBS and 1% antibiotics. Lapatinib medium was removed and cultured in drug-free medium until 80% viable cells grew. Then, the same concentration was repeatedly exposed to cells for 2 weeks. Cells were cultured every 2 weeks in medium containing an increased lapatinib concentration of 50 nM. Finally, resistant cell lines that grew exponentially in the presence of high concentrations were established as lapatinib-resistant breast cancer cell lines. The BT474 cell line was maintained at 1 μM, and the SKBR3 cell line was maintained at 0.6 μM. Before each experiment, resistant cell lines were maintained in medium without lapatinib for at least 2 or 3 days.

### 4.3. Western Blotting

Cells were lysed in PRO-PREPTM protein extraction solution (iNtRON Biotechnology, Cat# 17081) supplemented with phosphatase inhibitor cocktail (Gen DEPOT, Cat. No. p3200-001). Protein concentrations were then measured using a Bio-rad Bradford assay (Cat# 500-0006, Hercules, CA, USA). Equal amounts of protein were separated by SDS-polyacrylamide gel electrophoresis and transferred to PVDF membranes (Cat# 10600023). The membranes were blocked with Tris-buffered saline containing 0.05% Tween 20 and 5% non-fat dry milk for 1 h, and membranes were incubated overnight at 4 °C with primary antibody. HER2 (Cat# 2165, RRID:AB_10692490), p-HER2 (Cat# 2247, RRID:AB_331725), p-EGFR (Cat# 2234, RRID:AB_331701), Akt (Cat# 9272, RRID:AB_329827), p-Akt (Cat# 9271, RRID:AB_329825), ERK (Cat# 9102, RRID:AB_330744), and p-ERK (Cat# 9101, RRID:AB_331646) antibodies were obtained from Cell Signaling Technology (Beverly, MA, USA), EGFR (Cat# sc-03, RRID:AB_631420) was obtained from Santa Cruz, and ß-actin (Cat# A5316, RRID:AB_476743) was obtained from Sigma (Saint Louis, MO, USA). Then, horseradish peroxidase-conjugated secondary antibodies were incubated for 1 h at room temperature. Immunoreactive bands were detected with chemiluminescence reagent (Cat# RPN2106).

### 4.4. Microarray Analysis

Total RNA was extracted from SKBR3, LR-SKBR, BT474, and LR-BT474 cells using Trizol reagent (Invitrogen, Cat# 10296-010, Carlsbad, CA, USA) according to the manufacturer’s instructions. RNA was labeled, amplified, and hybridized to Illumina Human HT-12 v4 (48K) according to the Illumina standard protocol by an Agilent-certified service provider (Macrogen, Seoul, Korea). Total RNA amplification and data extraction were performed as previously published [45].

### 4.5. SILAC-Based Mass Spectrometry

For cell labeling, basal SILAC media without L-arginine and L-lysine were composed of 10% dialyzed FBS, antibiotic penicillin/streptomycin solution. Light media were supplemented with light lysine (K) and arginine (R) for LR-BT474 or LR-SKBR3, and heavy media of ^13^C_6_^15^N_2_-K and ^13^C_6_^15^N_4_-R (Cambridge Isotope Laboratories, Tewksbury, MA, USA) for BT474 or SKBR3. Cells were grown in light and heavy media for at least five passages. Phosphopeptide enrichments, mass spectrometry runs, and data analyses were performed as published previously [46].

### 4.6. Cell Viability Assay

SKBR3 and BT474 cells were seeded in a 6-well (2 × 10^5^ cells/well) plate with 2 mL of growth medium and treated with lapatinib the next day. After incubation for 48 and 72 h, cells were incubated with 100 μL of MTT solution for 4 h at 37 °C. The supernatant was removed, and insoluble formazan dye was solved by 100 μL DMSO. Absorbance was measured at 540 nm by a microplate reader. 

### 4.7. Xenograft Model and Efficacy Study

All animal studies were approved by the Korea University Animal Care and Use Committee (# KOREA-2019-0129). First, 17-β-estradiol 60-day release pellets (Innovative Research of America) were transplanted into the left flank of athymic nude Foxn1nu female mice (Envigo, MA, USA) 1 day before tumor inoculation. After that, 5 × 10^6^ of BT474 cells and 5 × 10^6^ of LR-BT474 cells mixed with 50% matrigel (Corning cat. 356231, New York, NY, USA) were subcutaneously implanted into the right flank of athymic nude mice. When tumors reached approximately 150 mm^3^, animals were randomized to treatment groups: vehicle, 17-DMAG, lapatinib, and combination of 17-DMAG and lapatinib (*n* = 4, group). 17-DMAG was administered by intraperitoneal injection three times per week for 5 weeks at 5 mg/kg. Lapatinib was administered for 5 weeks at 75 mg/kg by oral gavage daily during the study period. Using a digital caliper, the same investigator serially measured vertical tumor size twice a week. Tumor size was calculated according to the following formula: TV (mm 3) = (length [mm] × (width [mm] 2)/2. Animal weights were also measured twice a week.

### 4.8. Immunohistochemical Analysis of Xenografted Tumors

At the end of the study, tumors were fixed with 4% formalin provided by the Department of Pathology. In addition, Hematoxylin and Eosin (H&E) staining of tumor tissue and ER and HER2 staining were performed by a skilled pathology department. The HER2 and ER scores on completed slides were independently reviewed by pathologists.

### 4.9. Statistical Analysis

Graphs and statistical comparisons were performed in Graph-pad Prism v8.3 software. Multi T-test was used to compare lapatinib-resistant cell lines and parent cell lines for each drug. A two-way ANOVA with Bonferroni’s post-test was used to compare groups for mouse tumor growth. Additionally, a T-test was used to compare ER Allred scores between groups.

To investigate the combination effect of lapatinib and 17-DMAG in lapatinib-resistant cells, isobologram was plotted and combination index (CI) was calculated using CompuSyn (version 1.0) software (T. C. Chou and N. Martin, Memorial Sloan-Kettering Cancer Center, New York, NY, USA).

## 5. Conclusions

We identified HSP90 as a common node for acquired resistance to lapatinib in both BT474 and SKBR3 cells using proteomic analysis. However, in vitro and in vivo studies demonstrated synergy between lapatinib and an HSP90 inhibitor, 17-DMAG, in LR-BT474 cells only. Further GSEA using transcriptional analysis revealed that the two HER2 (+) breast cancer cell lines have different resistance mechanisms after long-term exposure to lapatinib; estrogen response-related genes are major mechanisms in BT474 cells. The molecular mechanisms we delineated here may be a potential strategy for future clinical trials using HSP90 inhibitors in treatment for refractory HER2 (+) metastatic cancer patients.

## Figures and Tables

**Figure 1 cancers-12-02630-f001:**
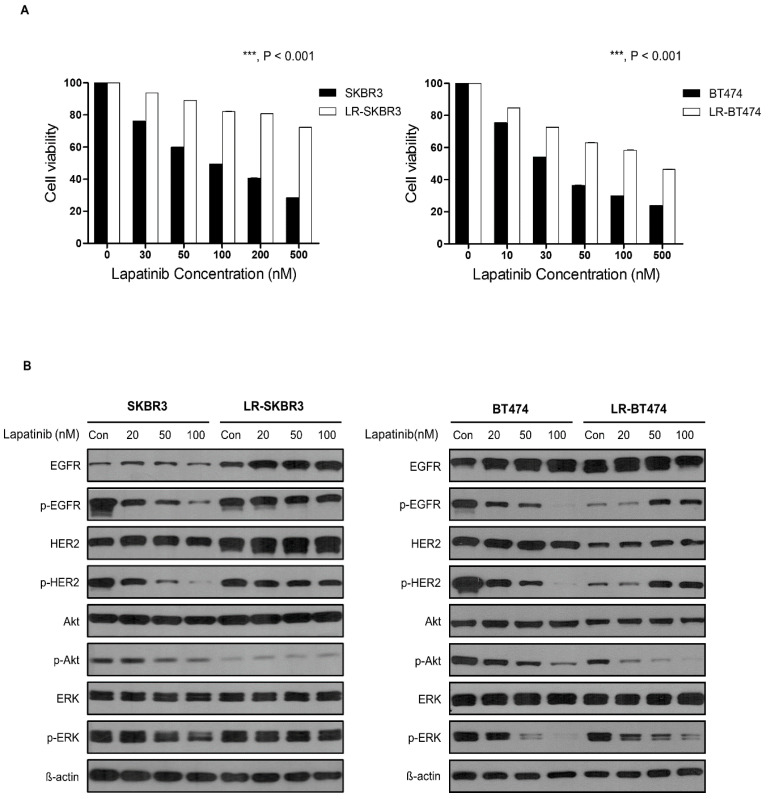
Establishment of lapatinib-acquired-resistant Human Epidermal growth factor Receptor 2 (HER2) (+) cell lines and biologic changes in HER2 downstream signaling. (**A**) Lapatinib was administered for 48 and 72 h at different concentrations in SKBR3, BT474, LR-SKBR3, and LR-BT474 cell lines and measured using MTT (3-[4,5-dimethylthiazol-2-yl]-2,5-diphenyltetrazolium) analysis. Results are expressed as percentage of viable cells from three independent experiments (mean ± SD) (***, *p* < 0.001). (**B**) Parental and lapatinib-resistant cell lines were treated with lapatinib at 20, 50, or 100 nM concentrations for 24 h. Western blot was performed using same amounts of protein. Protein activation was analyzed by evaluating phosphorylation status using corresponding p-HER2, p-EGFR, p-Akt, and p-Erk antibodies. Results were obtained from two independent experiments.

**Figure 2 cancers-12-02630-f002:**
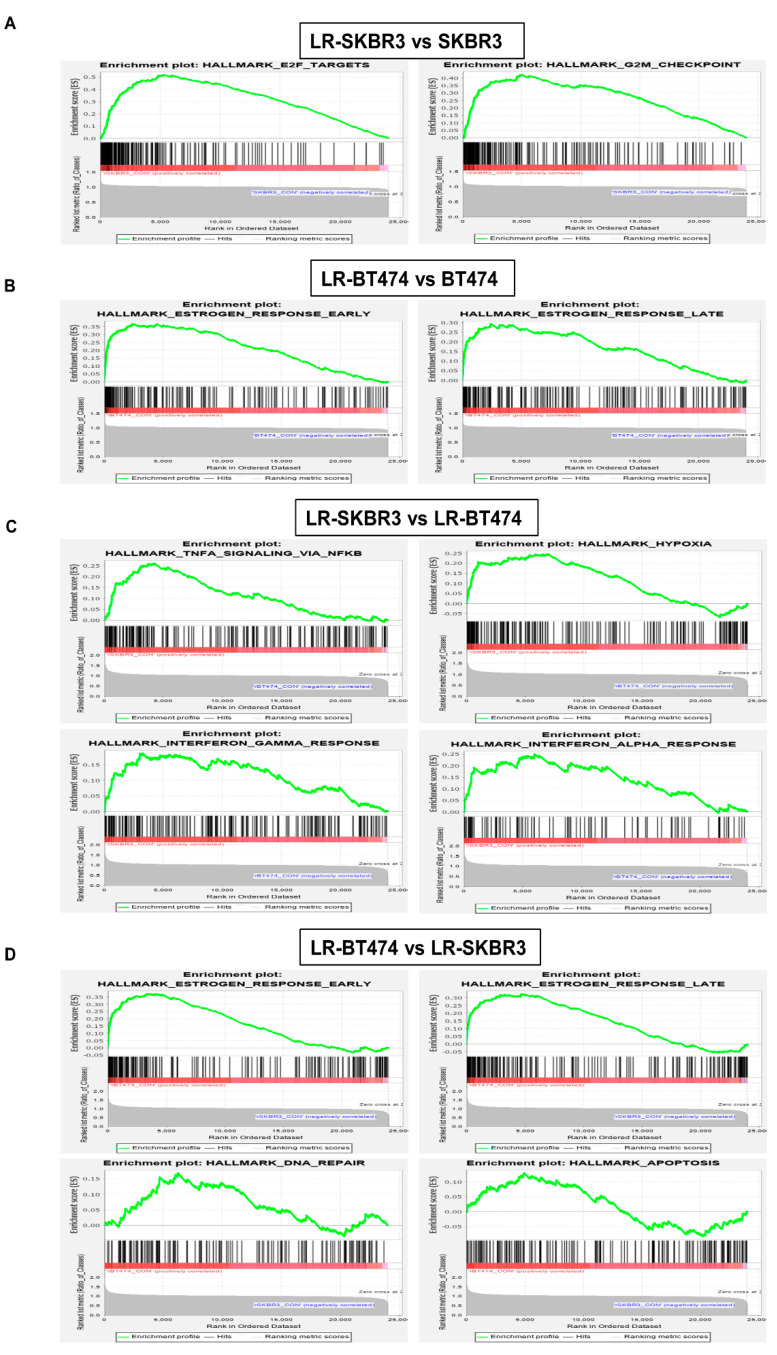
Gene set enrichment analysis (GSEA) enrichment plot in lapatinib-resistant cell lines. (**A**) GSEA plot shows G2M checkpoint and E2F target in LR-SKBR3 and SKBR3 cells. (**B**) GSEA plot shows early and late estrogen responses in LR-BT474 and BT474 cells. (**C**) GSEA analysis represents gene expression enrichment in LR-SKBR3 cell compared with LR-BT474 cell. (**D**) GSEA analysis represents gene expression enrichment in LR-BT474 cell compared with LR-SKBR3 cell.

**Figure 3 cancers-12-02630-f003:**
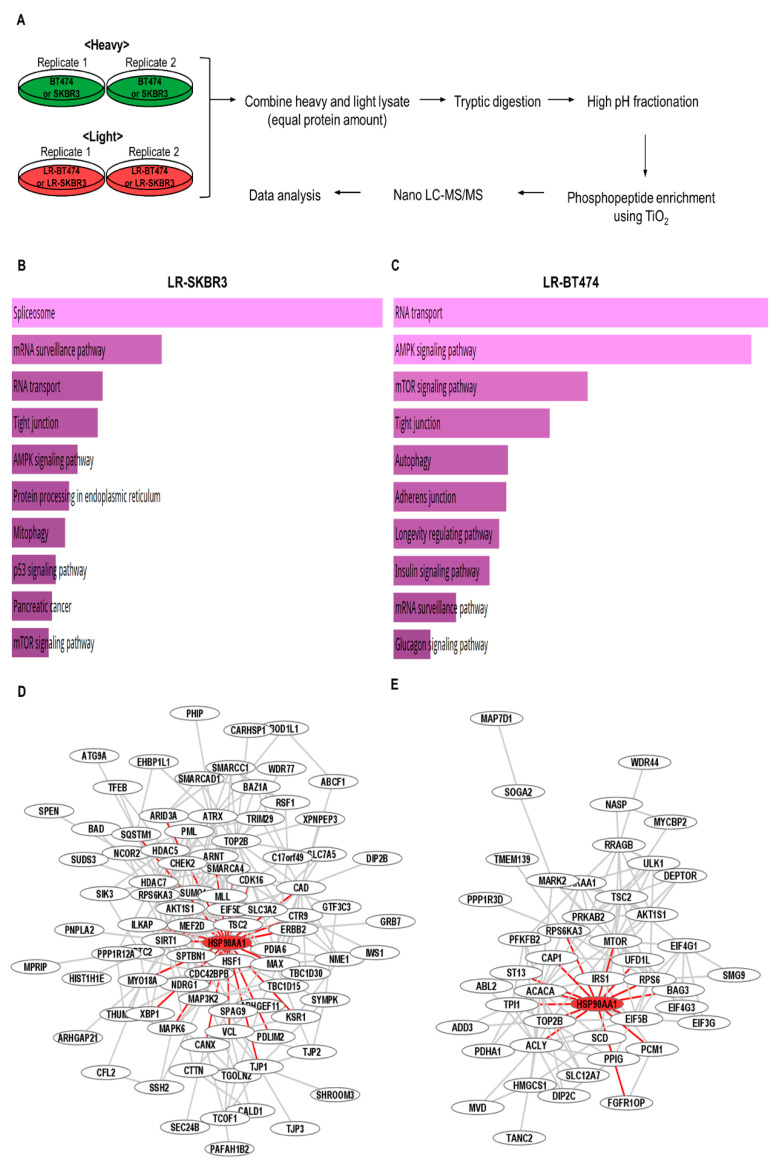
Phosphoproteomic profiling of lapatinib-resistant and sensitive cell lines. (**A**) Experimental schematic outline of SILAC experiment. (**B**) Enriched cellular pathways in LR-SKBR3. (**C**) Enriched cellular pathways in LR-BT474. (**D**) Major cluster by GLay clustering in LR-SKBR3. (**E**) Major cluster by GLay clustering in LR-BT474.

**Figure 4 cancers-12-02630-f004:**
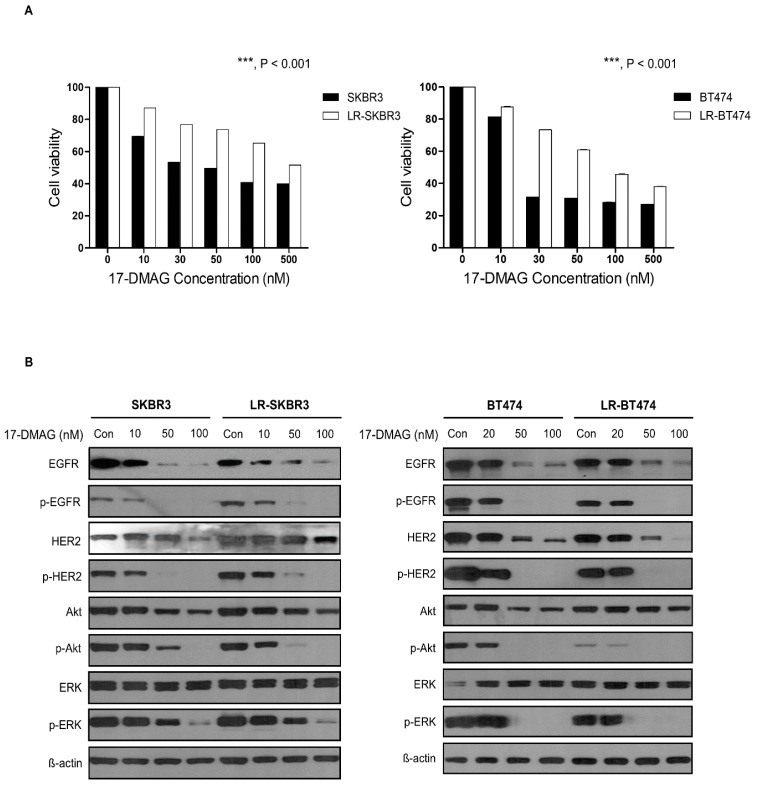
17DMAG induces inhibition of cell proliferation and affects HER2 downstream signaling in lapatinib-resistant cell lines. (**A**) 17DMAG was administered for 48 and 72 h at different concentrations in parent and lapatinib-resistant cell lines and measured using MTT analysis. Results are expressed as percentage of viable cells from three independent experiments (mean ± SD) (***, *p* < 0.001). (**B**) Parental and lapatinib-resistant cell lines were treated with 17DMAG at 20, 50, or 100 nM concentrations for 24 h. Western blot was performed using same amounts of protein. Protein activation was analyzed using corresponding p-HER2, p-EGFR, p-Akt, and p-ERK antibodies. Results represent 2 independent experiments.

**Figure 5 cancers-12-02630-f005:**
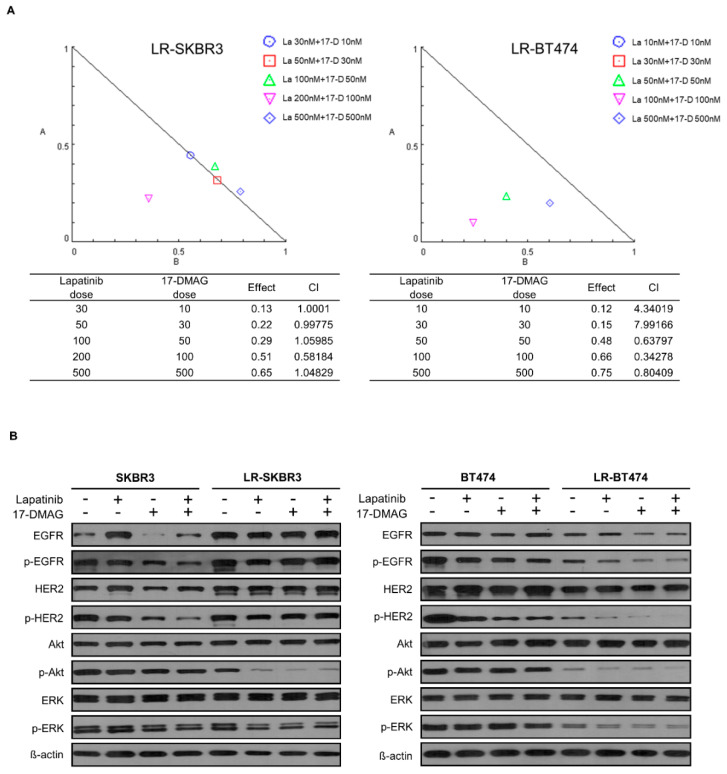
Combination of lapatinib and 17DMAG in lapatinib-resistant cell lines. (**A**) Isobologram plot shows combination treatment of lapatinib and 17-DMAG in LR-SKBR3 and LR-BT474 cell lines. (**B**) All cell lines were treated with lapatinib, 17DMAG, or combination lapatinib and 17DMAG for 24 h, lysed, and analyzed using western blot with indicated antibodies.

**Figure 6 cancers-12-02630-f006:**
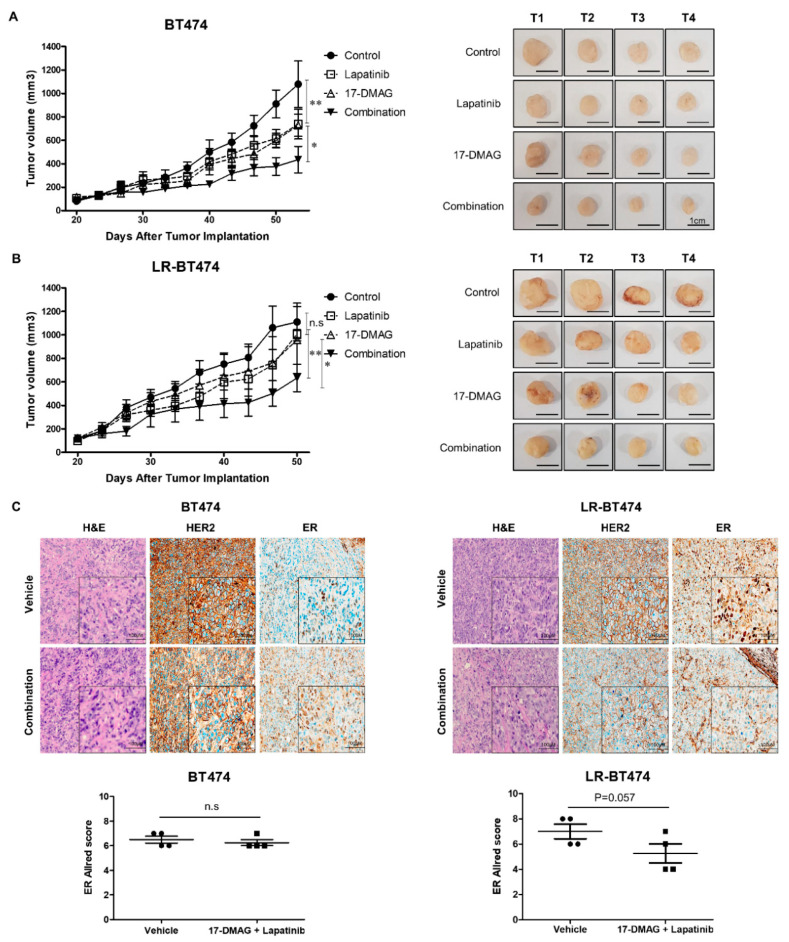
Antitumor efficacy of 17DMAG and lapatinib combination in BT474 and LR-BT474 xenograft models. (**A**,**B**) Mice were injected with BT474 and LR-BT474 cells on the flanks. After tumor formation, randomly grouped nude mice were treated with control, lapatinib (75 mg/kg), 17DMAG (5 mg/kg), or combination (lapatinib ,17DMAG) for 5 weeks. Tumor measurements were obtained twice a week. Average tumor volumes in control, lapatinib, 17-DMAG, and combination treatment groups are presented as mean ± SEM (*, *p* < 0.05; **, *p* < 0.01). Scale bars, 1 cm. (**C**) Representative image of immunohistochemical staining in BT474 and LR-BT474 tumors of xenograft models. Tumor sections from vehicle and combination groups were stained for HER2 and ER. Scale bars, 100 μm. Scatter plot of ER Allred scores in BT474 and LR-BT474 xenograft tumors measured by IHC. Data are presented as mean ± SEM (*n* = 4 in each group).

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
