# Peer review of "HSP90 Inhibitor, 17-DMAG, Alone and in Combination with Lapatinib Attenuates Acquired Lapatinib-Resistance in ER-positive, HER2-Overexpressing Breast Cancer Cell Line"

_cancers, 2020, doi:10.3390/cancers12092630_

Round 1
Reviewer 1 Report
The manuscript "HSP90 Inhibitor, 17-DMAG, Alone and in Combination With Lapatinib Attenuates Acquired Lapatinib-Resistance in HER2-Overexpressing Breast Cancer Cell Lines" submitted to Cancers for publication as a full article reported an interesting study about the resistance to lapatinib in breast cancer patients.
The manuscript is essential to understanc the role of lapatinib in breast cancer therapy and merits publication in Cancers. Although, some issues should be solved:
-the introdcution should be implemented for the readership of this journal. More data should emphasize the quality of the article.
-why MCF7 cells were not used as a typical control in breast cancer drug discovery?
-why the experiments were not conducted in breast tissues derived from female humans?
-please, use high resolution images for publication.
Congratulations for the work
Author Response
Reviewer 1
The manuscript "HSP90 Inhibitor, 17-DMAG, Alone and in Combination with Lapatinib Attenuates Acquired Lapatinib-Resistance in HER2-Overexpressing Breast Cancer Cell Lines" submitted to Cancers for publication as a full article reported an interesting study about the resistance to lapatinib in breast cancer patients.
The manuscript is essential to understand the role of lapatinib in breast cancer therapy and merits publication in Cancers. Although, some issues should be solved:
1.the introduction should be implemented for the readership of this journal. More data should emphasize the quality of the article.
- Thank you for your comments.
We implemented some data to give more insights about the disease that we have studied in introduction section (Line 42-44, 54).
- why MCF7 cells were not used as a typical control in breast cancer drug discovery?
- Yes, MCF-7 cells are one of the most frequently used in breast cancer studies. In this study, however, naturally HER-2 overexpressing cell lines (SKBR-3 and BT-474) were available and their biology are known to be stably maintained in long-term in vitro culture. Thus, we choose natural cell lines rather than MCF-7 cells with HER2 gene transfection. In our opinion, they are more suitable for study and investigate mechanism of lapatinib resistance with long-term exposure.
- why the experiments were not conducted in breast tissues derived from female humans?
- We appreciate for this comment. We also wanted to know if HSP90 is overexpressed on acquired resistance to lapatinib treatment in patients. However, the question can be answered by a study using paired tissue samples before and after the treatment which is very difficult in advanced cancer patients due to accessibility.
- please, use high resolution images for publication.
- Thanks for the valuable comment. The images were replaced by high resolution version (300dpi) and were uploaded again.
Congratulations for the work
Reviewer 2 Report
This manuscript describes a new potential combinational target, HSP90, for Lapatinib-resistant HER2 positive breast cancers. It’s very interesting to see that the two different cell lines show completely different resistance mechanisms. Combination of Lapatinib and HSP90 inhibitor only showed synergic effect in one of the cell lines. Some major and minor issues that can be improved are listed below:
Major issues:
- Please provide some background explaining how the network of EGFR, HER2, AKT and ERK (and their phosphorylated forms) works to help the audience understand why their levels are assessed and why their activation can cause the resistance. Is EGFR pathway commonly activated in HER2+ breast cancers?
- This sentence in the introduction “Currently, lapatinib has been approved for use as a combination with…” (48-51) is confusing. Why would use combination of lapatinb and capecitabine? And what is “chemotherapy using dual anti-HER2 therapy”?
- How long does it take SKBR3 and BT474 lines to become resistance, respectively? Is one faster than the other? Do you have data to show the gradual changes of their IC50s during the 1-year treatment?
- As the two resistant cells lines showed different alterations, is there any other known difference between the two lines, besides ER expression, that may explain the different resistance mechanisms?
- How were the two lines chosen for the study and any other cell lines were evaluated?
- What’s the bio-functions of HSP90 and how does it relate to HER2 and EGFR pathways? How does HSP90 contribute to the Lapatinib resistance in the two different mechanisms?
- The title of section 2.6 is inaccurate as it’s synergistic only in LR-BT474 cell line.
- How were the doses chosen for the treatments in mouse study, and any toxicity observed during the treatments?
- Does this study aim to use combination with HSP90 inhibitor for naïve or the resistant HER2+ breast cancers? It would provide more clinical significance if can show that using HSP90 inhibitor can prevent/slow down the onset of Lapatinib resistance.
Minor issues:
- What are “early/late estrogen response” (93-93)?
- In section 2.5, please briefly explain what is 17-DMAG and how does it work as a HSP90 inhibitor.
- The title should also point out that only HSP90 inhibitor only synergizes with lapatinib in ER+ cells.
Author Response
Reviewer 2
This manuscript describes a new potential combinational target, HSP90, for Lapatinib-resistant HER2 positive breast cancers. It’s very interesting to see that the two different cell lines show completely different resistance mechanisms. Combination of Lapatinib and HSP90 inhibitor only showed synergic effect in one of the cell lines. Some major and minor issues that can be improved are listed below:
Major issues:
1.Please provide some background explaining how the network of EGFR, HER2, AKT and ERK (and their phosphorylated forms) works to help the audience understand why their levels are assessed and why their activation can cause the resistance. Is EGFR pathway commonly activated in HER2+ breast cancers?
- Thanks for the valuable comments.
HER2/Neu (ErbB2) is a member of the ErbB family of transmembrane receptor tyrosine kinases, which also includes the epidermal growth factor receptor (EGFR, ErbB1), HER3 (ErbB3), and HER4 (ErbB4). In particular, co-expression of human epidermal growth factor receptor 2 (HER2) and EGFR has been reported to be associated with worse survival in HER2-positive breast cancer patients. Activation of the HER2 pathway leads to receptor autophosphorylation in C-terminal tyrosines and the recruitment to these sites of cytoplasmic signal transducers that regulate cellular processes such as proliferation, differentiation, motility, adhesion, protection from apoptosis, and transformation. Cytoplasmic signal transducers that are activated by this pathway include PLC-γ1, MAPK/Erk1/2 and PI3K/Akt , Src, the stress-activated protein kinases (SAPKs), PAK-JNKK-JNK, and the signal transducers and activators of transcription (STATs).)
2.This sentence in the introduction “Currently, lapatinib has been approved for use as a combination with…” (48-51) is confusing. Why would use combination of lapatinib and capecitabine? And what is “chemotherapy using dual anti-HER2 therapy”?
- Sorry for the unclear expressions.
We revised and clarified the sentences that you pointed out.
Currently, lapatinib has been approved for use as a combination with…
→ Currently, a combination of lapatinib and capecitabine has been approved for use in patients…
using dual anti-HER2 therapy
→ using dual anti-HER2 therapy; trastuzumab and pertuzumab
- How long does it take SKBR3 and BT474 lines to become resistance, respectively? Is one faster than the other? Do you have data to show the gradual changes of their IC50s during the 1-year treatment?
- Thanks for the valuable comment.
The time it took for the cells to gain resistance was similar for the two cells. In our experience, SKBR3 cell line was more difficult and trickier to establish a resistant cell line than BT474 cell line. Acquiring resistance was monitored by MTT assay. Please, refer the data at 4 month and 7-month time points.
- As the two resistant cells lines showed different alterations, is there any other known difference between the two lines, besides ER expression, that may explain the different resistance mechanisms?
- That’s a good point. One of the major differences between the two cell lines is mutation of PIK3CA. Some data support presence of mutated PIK3CA is one of the resistance mechanisms to HER-2 targeted agents. In terms of HSP90 targets, BT474 may have more deregulated targets for drug action. We will address that point in discussion section. Thank you very much!
- How were the two lines chosen for the study and any other cell lines were evaluated?
- Thanks for the valuable comment. To answer the questions that we had in this study, we looked for HER2 overexpressing cells. Among the more than 20 HER2 type and several HER2(+) ER (+) cell lines, we choose SKBR3 and BT474 cells as the two cell lines were one of the most extensively investigated cells and are easily available. Thus, the results from our study could be a reference for other researchers. For the subsequent study, we stared to establish other cell lines with different genetic background to reflect real patients.
- What’s the bio-functions of HSP90 and how does it relate to HER2 and EGFR pathways? How does HSP90 contribute to the Lapatinib resistance in the two different mechanisms?
- We agree that this is a very important point in this study. Thus, we already have described in the discussion section (Line 252-281). Please, refer the discussion section.
The HSP90 chaperone complex regulates the stability, activation, and maturation of more than 200 client proteins, including receptor tyrosine kinases (HER2, EGFR, IGF-1R, MET), signaling proteins (AKT, ERK) and cell cycle regulatory proteins. Although the mechanisms of lapatinib resistance acquired in the two cell lines differ, targeting HSP90 offers the potential to simultaneously destroy multiple carcinogenic pathways, as many proteins involved in alternative survival pathways are stabilized and activated by the HSP90 chaperone complex.
- The title of section 2.6 is inaccurate as it’s synergistic only in LR-BT474 cell line.
- Thanks for the valuable comment. We revised the title of the section as advised by the reviewer.
- How were the doses chosen for the treatments in mouse study, and any toxicity observed during the treatments?
- We choose the drug doses based on the previous research papers in vivo (doi: 10.18632/oncotarget.19375, doi:10.1158/1078-0432.CCR-07-1104). As there is a possibility of overlapping toxicity between the two combined drugs, less than the appropriate doses were used in this study.
Weight loss was expected from loss of appetite and diarrhea in case of toxicity, mice were weighed twice a week, but animals in combination treatment group showed little weight loss.
- Does this study aim to use combination with HSP90 inhibitor for naïve or the resistant HER2+ breast cancers? It would provide more clinical significance if can show that using HSP90 inhibitor can prevent/slow down the onset of Lapatinib resistance.
- We definitely agree with your opinion.
We aimed to evaluate if sub-optimal dose (less toxic) of HSP90 inhibitor would overcome acquired lapatinib resistance. As current standard of care with lapatinib is as a part of combination with capecitabine, we propose to use of HSP90 inhibitor in combination with HER2 target agent on acquired resistance, rather than in naïve setting. However, we are inspired to test HSP90 inhibitor in naïve setting to delay lapatinib resistance in the future animal study. Thank you very much!
Minor issues:
- What are “early/late estrogen response” (93-93)?
- Data from cDNA microarrays have shown that transcriptional response to 17β-Estradiol (E2) is different among the genes. Usually, differently expressed genes at 3-4 hrs (early) and at 24 hrs (late) were examined by pathway analysis. In a meta-analysis, 15 early response pathways, mostly related to cell signaling and proliferation, and 20 late response pathways, mostly related to breast cancer, cell division, DNA repair and recombination have been reported. [BMC Syst Biol. doi: 10.1186/1752-0509-5-138]
- In section 2.5, please briefly explain what is 17-DMAG and how does it work as a HSP90 inhibitor.
-Following the reviewer's advice, we briefly explained the role of HSP90 inhibitors, in sections 2.5, line 151-153.
- The title should also point out that only HSP90 inhibitor only synergizes with lapatinib in ER+ cells.
Following reviewer’s comment, we revised title of this manuscript.
HSP90 Inhibitor, 17-DMAG, Alone and in Combination with Lapatinib Attenuates Acquired Lapatinib-Resistance in ER-positive, HER2-Overexpressing Breast Cancer Cell Line
Reviewer 3 Report
IReviewer 3
In this manuscript the authors using in vitro cell lines and in vivo mouse tumor formation show that HSP90 inhibitor 17-DMAG synergizes with lapatinib in targeting breast cancer. Although the overall outcome of the study is not novel, with some modifications this can be a useful investigation.
- For Figure 1A, what was the time point? 48 or 72 hrs?
- Because the doubling times of the SKBR3 and BT474 cell lines are different, the experiment was performed with 48 hours for SKBR3 and 72 hours for BT474.
- The figure 3 is not legible. The font size needs to be increased.
- Thank you for the comment.
We replaced with a data image with a better resolution.
- For figure 4B, provide a lower exposure blot for HER2
- We apologize for the high background. Unfortunately, original membrane is not available, but we have a developed film only. Instead of low exposed data, color correction was done again and was inserted into the data.
- For the synergy studies, elaborate how the experiment was done and inferred. Show the results of MTT assay when combination was used.
We agree with you in that point. Following information was implemented in the results section (line 178-180). Synergy studies were done with the doses of lapatinib and 17-DMAG determined in the MTT assays using single drug (Fig 1 and Fig 4). Sub-IC50 doses in each parent cells were used for initial combination.
- Out of 2 cell lines tried, only one showed synergy. It would be necessary to show at least one more cell line.
- Thank you for your valuable comments. It would be great to add one more cell line with similar biology with BT474. However, it takes one year to establish lapatinib resistant cell line in the lab.
We admit that presenting only one cell line data is a limitation. However, please understand that this study focused on the difference in the mechanism of resistance to lapatinib and effect of HSP90 according to co-expression of ER in HER2-positive breast cancer cell lines.
- Recheck the title for the figure legend 5 (GSEA).
- We revised the legend as advised by the reviewer.
- The manuscript needs to be rechecked for English, syntax and sentence construction. Some places include capital C in line 97, line 109 (GSEA plots show).
- Thank you for the comments.
The manuscript was edited by a native English editor already.
We corrected typos and uploaded revised GSEA plots.
Author Response
Reviewer 3
In this manuscript the authors using in vitro cell lines and in vivo mouse tumor formation show that HSP90 inhibitor 17-DMAG synergizes with lapatinib in targeting breast cancer. Although the overall outcome of the study is not novel, with some modifications this can be a useful investigation.
- For Figure 1A, what was the time point? 48 or 72 hrs?
- Because the doubling times of the SKBR3 and BT474 cell lines are different, the experiment was performed with 48 hours for SKBR3 and 72 hours for BT474.
- The figure 3 is not legible. The font size needs to be increased.
- Thank you for the comment.
We replaced with a data image with a better resolution.
- For figure 4B, provide a lower exposure blot for HER2
- We apologize for the high background. Unfortunately, original membrane is not available, but we have a developed film only. Instead of low exposed data, color correction was done again and was inserted into the data.
- For the synergy studies, elaborate how the experiment was done and inferred. Show the results of MTT assay when combination was used.
We agree with you in that point. Following information was implemented in the results section (line 178-180). Synergy studies were done with the doses of lapatinib and 17-DMAG determined in the MTT assays using single drug (Fig 1 and Fig 4). Sub-IC50 doses in each parent cells were used for initial combination.
- Out of 2 cell lines tried, only one showed synergy. It would be necessary to show at least one more cell line.
- Thank you for your valuable comments. It would be great to add one more cell line with similar biology with BT474. However, it takes one year to establish lapatinib resistant cell line in the lab.
We admit that presenting only one cell line data is a limitation. However, please understand that this study focused on the difference in the mechanism of resistance to lapatinib and effect of HSP90 according to co-expression of ER in HER2-positive breast cancer cell lines.
- Recheck the title for the figure legend 5 (GSEA).
- We revised the legend as advised by the reviewer.
- The manuscript needs to be rechecked for English, syntax and sentence construction. Some places include capital C in line 97, line 109 (GSEA plots show).
- Thank you for the comments.
The manuscript was edited by a native English editor already.
We corrected typos and uploaded revised GSEA plots.
Round 2
Reviewer 2 Report
Thank you for the detailed responses to all the questions/issues. These are very helpful. However, there are still a few major issues that need to be further addressed.
For example, for the major issue 1, it would be important to provide information in the manuscript to briefly explain why you chose the levels of AKT, ERK etc as markers in your experiments, to help the audience from different background to understand why you are doing this.
#4. This is one of the key questions, but I did not see that explanation in the revised discussion.
#6. It has been explained that HSP90 regulates many oncogenic factors including HER2, EGFR and Akt pathways, but didn’t include any details how, and how HSP90 contributes to the two different resistant mechanisms. Especially, please explain and focus on HSP90A’s function in this and how it differs from the other member of HSP90 family, HSP90B. Additionally, the discussion did not connect the different resistant mechanisms to the synergism with HSP90 inhibitor, as synergy was only seen in BT474 line which is ER+, “early/late estrogen response gene” enriched, and “RNA transport, and AMPK signaling pathway, mTOR signaling pathway” enriched, which are different from the other line. How do these altered genes/pathways relate to HSP90?
Minors:
Line 98-98: As mentioned, please provide explanation for “early/late estrogen response” in the manuscript and what does the enrichment of these genes indicate, how it’s involved in the resistance.
Line 155: please change to “treatment with a HSP90 inhibitor, 17-dimethylaminoethylamino-17-demethoxygeldanamycin (17-DMAG)”
Line 242: please add “(TKIs)” after tyrosine kinase inhibitors, so that abbreviation “TKI” can be used after.
Line 266: please define docetaxel
Author Response
Thank you for the detailed responses to all the questions/issues. These are very helpful. However, there are still a few major issues that need to be further addressed.
For example, for the major issue 1, it would be important to provide information in the manuscript to briefly explain why you chose the levels of AKT, ERK etc as markers in your experiments, to help the audience from different background to understand why you are doing this.
- Following the reviewer's advice, we briefly explained the relevance of the downstream signals in the introduction section, line 40-48.
#4. This is one of the key questions, but I did not see that explanation in the revised discussion.
- Thanks for the valuable comment. We included the relevant explanation in the discussion section (Line 327-332). We apologize for the mistake.
#6. It has been explained that HSP90 regulates many oncogenic factors including HER2, EGFR and Akt pathways, but didn’t include any details how, and how HSP90 contributes to the two different resistant mechanisms. Especially, please explain and focus on HSP90A’s function in this and how it differs from the other member of HSP90 family, HSP90B.
- There are two main mammalian isoforms of HSP90, presumably products of gene duplication, localized in the cytoplasm; the inducible form Hsp90α and the constitutive form HSP90β. Thus, Hsp90α and constitutive form HSP90β proteins have 85% sequence identity. Although these cytoplasmic HSP90 isoforms have recently shown distinctive functions, most reviews do not distinguish them due to their remarkable structural and functional similarity. Therefore, the name HSP90 has been used for both HSP90 α and β unless indicated. Studies dealing with the differentiation between cytoplasmic HSP90 isoforms, revealed that although both isoforms very often form dimers, HSP90α tends to dimerize frequently compared to HSP90β. (doi: 10.3390/ijms19092560)
To clarify the context of the paper, we added following to the discussion section (Line 264-270).
‘The HSP90s are highly conserved molecular chaperone proteins as one of the sub-family of HSPs and have distinctive role in stability and function of conformationally labile proteins. In cancer cells, HSP90 expression is induced by microenvironmental stresses, including hypoxia, chemotherapy, and immunologic attack, rather than gene amplification or point mutation. It’s chaperone complex regulates the stability, activation, and maturation of more than 400 client proteins, including oncogenic receptor tyrosine kinases (HER2, EGFR, IGF-1R, MET), signaling proteins (AKT, ERK) and cell cycle regulatory proteins.’
Additionally, the discussion did not connect the different resistant mechanisms to the synergism with HSP90 inhibitor, as synergy was only seen in BT474 line which is ER+, “early/late estrogen response gene” enriched, and “RNA transport, and AMPK signaling pathway, mTOR signaling pathway” enriched, which are different from the other line. How do these altered genes/pathways relate to HSP90?
- Thank you for your valuable comments! We added more interpretation on that point in the discussion section as following. [line 327-332]
‘Given the data from GSEA and IHC analysis of the in vivo experiment, down-regulation of ER expression in LR-BT474 cells might be an important role in the synergistic effect of 17-DMAG. As for TNFα, interferon γ, interferon α, and hypoxia, more enriched resistant mechanisms in the LR-SKBR3 cells does not seem to have much effect by 17-DMAG.’
Minors:
Line 98-98: As mentioned, please provide explanation for “early/late estrogen response” in the manuscript and what does the enrichment of these genes indicate, how it’s involved in the resistance.
-Following the reviewer's advice, we briefly explained the early/late estrogen response, in sections 2.2, line 116-118. However, it seems difficult to explain how it is involved in resistance here, since only gene expression changes at the transcription level were confirmed in both cell lines.
Line 155: please change to “treatment with a HSP90 inhibitor, 17-dimethylaminoethylamino-17-demethoxygeldanamycin (17-DMAG)”
- Thanks for your valuable comments. Following the reviewer's advice, I revised in section 2.5, line 164.
Line 242: please add “(TKIs)” after tyrosine kinase inhibitors, so that abbreviation “TKI” can be used after.
- Thanks for your valuable comments. Following the reviewer's advice, I revised line 250 in the discussion section
Line 266: please define docetaxel
- Following the reviewer's advice, we briefly explained in the discussion sections, line 278-279.
